# Morphophysiological and Nutritional Responses of Bean Cultivars in Competition with *Digitaria insularis*

**DOI:** 10.3390/plants14172684

**Published:** 2025-08-28

**Authors:** Leandro Galon, Carlos Daniel Balla, Otilo Daniel Henz Neto, Lucas Tedesco, Germani Concenço, Ândrea Machado Pereira Franco, Aline Diovana Ribeiro dos Anjos, Otávio Augusto Dassoler, Michelangelo Muzell Trezzi, Gismael Francisco Perin

**Affiliations:** 1Laboratory of Sustainable Management of Agricultural Systems, Federal University of Fronteira Sul, Erechim Campus, Erechim 99700-000, Rio Grande do Sul, Brazil; otilohenz@gmail.com (O.D.H.N.); lcstede7@gmail.com (L.T.); andrea.franco@uffs.edu.br (Â.M.P.F.); anjosaline488@gmail.com (A.D.R.d.A.); odassoler@yahoo.com (O.A.D.); gismaelperin@gmail.com (G.F.P.); 2Post Graduation Program in Environmental Science and Technology of the Federal University of the Southern Border, Erechim Campus, Erechim 99700-000, Rio Grande do Sul, Brazil; d42balla@gmail.com; 3Laboratory of Sustainable Cropping Systems, Embrapa Temperate Agriculture, Pelotas 96010-971, Rio Grande do Sul, Brazil; germani.concenco@embrapa.br; 4Laboratory of Weed Science Research Centre, Federal University of Technology Paraná, Pato Branco Campus, Pato Branco 85503-390, Paraná, Brazil; trezzim@gmail.com

**Keywords:** *Phaseolus vulgaris*, competitive interaction, sourgrass, replacement series, plant interference

## Abstract

Studies exploring the competitive interactions between common beans and weeds are essential to adopt more efficient management strategies in the field, thereby reducing production costs. This study aimed to evaluate the competitive ability of bean cultivars in the presence of sourgrass (*Digitaria insularis*), using different plant proportions in associations. The experiments were conducted in a greenhouse, arranged in a randomized block design with four replications, from October 2020 to February 2021. Treatments were organized in the following plant proportions of beans and sourgrass: 100:0, 75:25, 50:50, 25:75, and 0:100%. The competitiveness analysis was carried out using replacement series diagrams and relative competitiveness indices. At 50 days after emergence (DAE), measurements were taken for leaf area, plant height, gas exchange, shoot dry mass, and nutrient concentration in bean leaves. The results show that interference between common bean cultivars and sourgrass involves equivalent competition mechanisms. Increasing sourgrass density negatively affects physiological traits and gas exchange in beans by about 10%. Beans show about 15% higher relative growth than sourgrass, based on competitiveness indices. Nutrient levels vary by cultivar and competitor ratio. Intercropping harms species more than intraspecific competition. Further field studies should determine critical control stages and economic impacts, aiding weed management decisions in bean production.

## 1. Introduction

Brazil planted approximately 2.86 million hectares of beans in the 2023/24 crop season, with an average yield of 1.14 t ha^−1^ [1], which is significantly lower than the yields from high-tech farms or research areas. To achieve the maximum yield potential of the crop, limiting factors such as weed competition must be controlled [2,3,4].

Weeds, by competing with crops, can deplete large amounts of nutrients and water from the soil, as well as other environmental resources such as light, further reducing availability to the cultivated plants [2,5,6]. Weeds may also release allelopathic compounds and host diseases and pests. These may cause reduced yields, depreciation of harvested grains, and harvesting difficulties, whether mechanical or manual [6,7,8].

Among the main weeds infesting bean crops, sourgrass (*Digitaria insularis*) stands out. This monocot species of the Poaceae family uses a C_4_ photosynthetic pathway, which provides greater efficiency in utilizing environmental resources such as light, water, and nutrients [4,9]. Sourgrass thrives in high temperatures, harbors pests that are harmful to crops, and is a perennial species capable of reproducing by seeds or rhizomes, even in low-fertility soils [4,9].

Integrated weed management is recommended for bean producers to reduce their reliance on herbicides, using them as a complementary and preferably last-resort tool [10]. Cultivar selection can be a strategy in weed management, as cultivars vary in their competitive ability due to genetic traits such as growth habit, plant height, architecture, leaf area index, root volume, branching or tillering ability, and biomass production, among others [10,11,12]. Some studies have reported sourgrass resistance to EPSPs-inhibiting herbicides (glyphosate) and ACCase inhibitors, commonly used to control this weed [13,14], during both desiccation before planting and post-emergence weed control.

Plants’ growth and development require macro- and micronutrients to perform photosynthesis, absorb light, and convert it into energy [15]. The most demanded nutrients—nitrogen, phosphorus, potassium, and carbon—are critical for yield, and their deficiency can lead to severe yield and grain quality losses [16]. These nutrients have essential functions: carbon contributes to biomolecular structures, nitrogen is part of macromolecules like nucleic acids, phosphorus is vital in energy processes, and potassium plays roles in photosynthesis, stomatal regulation, and water absorption [15].

A species with higher capacity to absorb and assimilate available resources in a given niche has greater development potential, thus showing superior competitiveness over neighboring species [12,17,18]. To maximize yields, crops must be adequately nourished and weeds properly managed to avoid competition for essential resources.

Determining crop–weed competitive interactions requires appropriate experimental designs and analysis methods. The replacement series method is widely used to investigate plants’ competitive ability [17], allowing for differentiation between intra- and interspecific competition [18]. Evaluating beans’ competitiveness against sourgrass using this method helps understand the influence of competition on both species and guides efficient management practices to avoid productivity losses [12,18].

This study hypothesizes that sourgrass adapts better and has greater competitive ability than beans when planted in equal proportions, using environmental resources more efficiently. Therefore, this work aimed to evaluate the relative competitive ability of bean cultivars in the presence of sourgrass, using different plant proportions in the associations.

## 2. Results and Discussion

The analysis of variance of the data showed a significant effect among the plant proportions of each common bean cultivar when competing with sourgrass for plant height (PH), leaf area (LA), chlorophyll index (CI), sub-stomatal CO_2_ concentration (Ci), stomatal conductance (Gs), transpiration rate (E), photosynthetic rate (A), carboxylation efficiency (CE), water-use efficiency (WUE), shoot dry mass (DM), and nutrient contents (N, P, K, and C).

### 2.1. Morphological Variables

The graphical results show that, for the combinations of common bean cultivars (BRS Esteio, IPR Uirapuru, IPR Urutau, BRS Estilo, IAC 1850, and IPR Tangará) with the sourgrass biotype (competitor), all six cultivars exhibited similarities when competing with the weed species. Significant differences were observed for plant height (PH), leaf area (LA), and dry mass (DM) in almost all plant proportions (Figure 1, Figure 2 and Figure 3; Table 1).

The replacement series experiments between the common bean cultivars versus sourgrass indicated competition between the crops and the weed, with the productivity values obtained in the different proportions between the two species generally deviating from the expected yield lines (PR and PRT).

In general, the association between the two species resulted in a mutual disadvantage for both the common beans and the sourgrass (Figure 1, Figure 2 and Figure 3; Table 1, Table 2, Table 3 and Table 4). A similar result was obtained by Gasparetto et al. [19], where both common beans and horseweed (*Conyza bonariensis*) exhibited negative effects on morphological variables when the species competed against each other.

Regarding PRT, significant differences were observed between the expected and estimated values in at least two proportions for plant height (PH), leaf area (LA), and dry mass (DM) for the common bean cultivars when competing with sourgrass. The lines were concave, and the average values were lower than 1 in all combinations, except for the cultivar IPR Uirapuru in the 75:25 proportion, where the PRT showed a value greater than 1 (Figure 1, Figure 2 and Figure 3; Table 1).

It was also noted that, for PH, the observed and estimated values were very close, unlike the PRT for LA and DM, where there was a greater distance between these values. The occurrence of etiolation in the common bean plants may explain this result, as the PRs for PH (Figure 1) remained above the dashed lines (expected values), especially up to the 50:50 proportion between common beans and sourgrass. However, this behavior was not reflected in the increased PRs for LA (Figure 2) or higher DM accumulation (Figure 3). The straight or slightly convex lines of the PRs for PH presented by the common bean cultivars may be attributed to competition for light, as evidenced by the greater stem elongation when in competition with the weed, which suppresses the development of LA and DM [20].

The estimated PRT for LA and DM showed significant differences, with the presence of concave lines and average values lower than 1 in all plant mixtures. This indicates competition for the same environmental resources, impairing the development of both the crop and the competitor. These losses were observed even at the lowest proportions of the weed, demonstrating that this species can cause damage to the crop even at low density (Figure 2 and Figure 3; Table 1). Other weeds, such as *Euphorbia heterophylla*, *Amaranthus* spp., *Bidens pilosa*, and *Urochloa plantaginea* [3,5,12,15,21], have also been reported to possess greater competitive ability in relation to common bean plants by negatively interfering with their growth and development.

In most situations, significant differences were reported in at least two plant proportions for the studied and expected PRs of common beans when competing with sourgrass for plant height (PH), leaf area (LA), and dry mass (DM) (Figure 1, Figure 2 and Figure 3; Table 1). Among the evaluated variables, LA and DM experienced the greatest reductions in the PR curve compared to the PH of both the crop and the competitor (Figure 1, Figure 2 and Figure 3). The least damage from the interaction in relation to PH may be associated with the plant’s strategy to improve its capture of light energy, which leads to the formation of longer stems or culms, with less energy investment in the development of LA or DM [20]. It is important to note that light is one of the most limited resources in communities and plays a crucial role in the initial response of a plant with greater competitive potential [22]. A similar result was observed in common beans’ competition with *A.* spp., *B. pilosa*, and *U. plantaginea*, where the crop was negatively affected in the presence of the weeds [12,15,21].

The relative PRs for leaf area (LA) (Figure 2) and dry mass (DM) (Figure 3), both for the crop and the competitor, were all represented by concave and significant lines (Table 1), except for the BRS Estilo cultivar, where no significance was observed for the PR of LA for the common bean. This demonstrates that the crop and the weed compete for the same resources in the environment, leading to mutual detriment in the growth of both species [23]. These losses are observed even with the lower proportions of the weed species, indicating that it can cause damage to the crop even at low densities. Common bean cultivars in competition with species like *U. plantaginea* [24], *C. bonariensis* [19], and *A.* spp. [12] also exhibited the occurrence of concave lines for both the crop and the competitor for the evaluated variables, which supports the results reported in this study.

The relative growth of the common bean cultivars was generally greater compared to the growth of sourgrass, regardless of the plant proportion evaluated (Figure 1, Figure 2 and Figure 3; Table 1). When specifically evaluating the 50:50 proportion between the species, which is considered to be a critical point for replacement series experiments, it was observed that all of the common bean cultivars exhibited higher PR values for plant height (PH), leaf area (LA), and dry mass (DM) (Table 1) compared to sourgrass. This suggests that the competitor has low competitive ability relative to the crop. The higher growth of the common bean may be a result of the crop having a faster initial development compared to sourgrass or showing greater leaf area and biomass accumulation, which could contribute to its superior competitive performance. Other studies have also reported that common beans demonstrated greater growth when in competition with different weed species, such as *U. plantaginea* [7,24], *C. bonariensis* [19], and *A.* spp. [12], corroborating the results reported in the present study. It can also be noted that weeds often appear in agricultural areas at densities higher than those of the cultivated plants, and in most situations they are considered to be more competitive in utilizing available environmental resources [17,19,23].

In general, it was observed that the common bean cultivars exhibited lower PR loss compared to sourgrass, regardless of the plant proportion in the association (Table 1). It was also noted that the total relative productivity (PRT) increased with the proportion of common bean plants, a situation that was significant for all of the variables studied. This behavior indicates that the species are competitive and that neither species contributes more than expected to the total productivity of the other [22]. It is worth highlighting that sourgrass (a weed species) has a slow initial growth, and its mass increase occurs after the first 45 days post-emergence, which is a period in which it causes significant competition when infesting the common bean, as reported by Barroso et al. [25]. This situation is partially similar to the results found in this study.

The morphological variables—plant height (PH), leaf area (LA), and dry mass (DM)—of the common bean cultivars were generally reduced when in competition with sourgrass, regardless of the plant proportion in the association (Table 2). The higher the proportion of the competitor in the association, the greater the morphological damage to the crop. For sourgrass plants, more pronounced reductions in PH, LA, and DM were observed when in equal or lower plant proportions compared to the common bean cultivars (Table 2). Several studies report growth losses for both the crop and the weed when in competition [5,12,18,25,26].

The lower values for LA and DM indicate high interspecific competition, where the species are competing for the same environmental resources (Table 2). Interspecific competition was also reported in studies involving common bean cultivars and *U. plantaginea* [27] and *C. bonariensis* [28]. It is possible that when the crop is well distributed, its competitive ability increases, while in row distribution—commonly used in the field—the damage caused by the weed community tends to increase [26].

In general, PH, LA, and DM showed the highest average values per plant for the species present in higher proportions in the association, regardless of the plant proportion (Table 2). It was observed that interspecific competition is more detrimental to both species involved in the study than intraspecific competition. These data are consistent with the findings of Dusabumuremyi et al. [26] and Gasparetto et al. [19], who evaluated common bean cultivars infested with weeds.

The reduction in the growth of species involved in either intraspecific or interspecific combinations is attributed to spatial competition between plant groups occupying the same niche [12,17,18,21]. However, greater interspecific competition is not limited to common bean cultivars competing with weeds. Several other studies involving plant species in competition have also found similar effects to those observed in the present study, such as rice and soybeans in the presence of *D. ciliaris* [18], or common beans infested by species such as *U. plantaginea* [24], *C. bonariensis* [19], and *A.* spp. [12].

### 2.2. Physiological Variables

The results demonstrate that the chlorophyll index of common bean cultivars was higher with a decrease in the proportion of bean plants and an increase in competition from sourgrass (Table 3). In common bean crops competing with *Commelina diffusa* (spreading dayflower), coexistence with the weed significantly reduced the chlorophyll content, number of pods, number of grains, grain yield, and nitrogen content in the seeds [27].

For the sub-stomatal CO_2_ concentration (Ci—µmol mol^−1^), only the cultivar IPR Urutau showed a lower value when not competing with any proportion of sourgrass (Table 3). The other cultivars presented higher values when the proportion was 100:0 (crop vs. competitor). This may be related to the stress caused by the weed, as Ci is a physiological variable influenced by environmental factors such as water availability, light, energy, and others [28]. Thus, the results found in this study are linked not only to competition for resources between species but also to environmental conditions.

All of the bean cultivars showed a reduction in stomatal conductance (Gs) when competing with the weed, compared to monoculture (Table 3). This is because the sourgrass caused stress to the crop. According to Tomazini et al. [4], when a plant is under stress (whether water, light, or nutritional stress) caused by interference from weeds, it adjusts its photosynthetic process according to the environmental conditions, and it may tend to close its stomata. This results in a reduction in gas exchange and the photosynthetic rate of the crop in response to available radiation.

Upon analyzing the gas exchange parameters, we observed an increase in photosynthetic activity (A) for all bean cultivars, except for IPR Uirapuru and IPR Urutau, which showed a reduction in this variable as the competition density of the weed increased (Table 3). The reduction in A was accompanied by a decrease in Gs and Ci, indicating stomatal limitation of photosynthesis, resulting from competition [29]. However, at the 50:50 or 25:75 (bean/sourgrass) ratio, despite changes in stomatal conductance, the A of the bean plants was maintained at levels similar to, or slightly higher than, the control without competition. In fact, A increased in the cultivars BRS Esteio, BRS Estilo, IPR Tangará, and IAC 1850 at the 25:75 (crop/weed) ratio, likely due to an increase in water-use efficiency (WUE).

The bean plant reacts to environmental stress by adjusting the opening and closing of its stomata, not only in response to solar radiation but, more importantly, in relation to the soil’s water potential, which can impact Ci as it is captured in smaller quantities [3,21]. Therefore, changes in the gas exchange of bean cultivars in competition with weeds result in both qualitative and quantitative effects on production. The efficiency of resource utilization (such as water, light, CO_2_, and nutrients) directly impacts the photosynthetic rate, water-use efficiency, growth, and productivity of the plants in the area [17,28].

For all bean cultivars, an increase in the transpiration rate (E) was observed as the density of the competitor plants infested the crop, when compared to the weed-free control (Table 3). This happened due to the competition between the crop and the weed for available resources such as water, nutrients, light, CO_2_, and growth space, which limits the growth of the crop, primarily affecting agricultural production [30]. These authors also argue that the increased competition for water by weeds in a field will exacerbate conditions of turgor loss, stomatal closure, and decreased photosynthesis and transpiration, interrupting cell growth and metabolic processes.

The carboxylation efficiency (CE—mol CO_2_ m^−2^ s^−1^) of bean plants in all cultivars increased with the increase in sourgrass densities competing with the crop, except for IPR Uirapuru, where no statistical significance was observed between the weed and the crop (Table 3). This occurs because this cultivar likely exhibits greater tolerance to sourgrass infestation, with no metabolic changes related to light absorption. Manabe et al. [21], when studying common beans in competition with weeds, observed a reduction in CE, stating that it was directly linked to the reduction in internal CO_2_ concentration and the decrease in assimilation rate.

The WUE is defined as the amount of water transpired by a species for biomass production, and it is greatly determined by the duration of stomatal opening [21]. In the present study, the WUE was reduced for most cultivars when competing with sourgrass (Table 3). For the WUE of black beans in competition with weeds, Manabe et al. [21] reported altered water absorption between species, with the crop consuming less water. The same authors also reported that the non-infested control consumed more water during the same period, demonstrating that the weed affects the crop, corroborating the results of the present study. According to Pessôa et al. [2], the physiological performance of common beans in the presence of weeds was negatively affected, with a reduction in the photosynthesis rate and an increase in transpiration rate and intercellular CO_2_ concentration.

### 2.3. Competition for Nutrients

Competition of the common bean cultivars IPR Urutau and BRS Estilo with sourgrass, regardless of plant densities in the association, showed no statistical significance for nutrient concentrations—nitrogen (N), phosphorus (P), and potassium (K)—meaning that there was no effect of competition with the weed (Table 4).

The common bean cultivars IAC 1850 and IPR Tangará showed no significant competition effects between the weed and the crop for P and K, respectively. For carbon (C), only IPR Urutau and IAC 1850 showed no effect of competitor densities. The base fertilization at sowing, combined with highly competitive common bean phenotypes, is essential for nutrient provision for plant growth and development [31].

The results for the cultivar BRS Esteio (Table 4), in the 50:50 proportion, demonstrate that there was an increase in the levels of N, P, and K compared to monoculture (control). In contrast, for the cultivar IPR Uirapuru, the opposite was observed, meaning that lower concentrations of these same nutrients were found in the same plant proportion (crop/weed). This may provide strong evidence of the higher and lower nutrient requirements for N, P, and K by the cultivars BRS Esteio and IPR Uirapuru, respectively, when competing with sourgrass, likely related to the specific genetic characteristics of each cultivar [10,12,32].

There were higher concentrations of N and P in the cultivar IPR Tangará when infested in the proportions of 50:50 and 25:75 compared to monoculture (100:0) (crop × competitor, respectively) (Table 4). In the same cultivar, for C, the opposite effect was observed, meaning that the higher the density of sourgrass plants, the lower the nutrient accumulation in the crop. When in competition, species focus photoassimilate use to the development of the aboveground parts, promoting stem elongation (etiolation) as a response to avoid shading and ensure greater light acquisition [20].

At the proportions of 75:25 and 25:75 (crop/weed), it was observed that the N and K contents of the cultivar IAC 1850 were higher for the first nutrient and lower for the second, when compared to the competition-free control (Table 4). This occurs due to the plant’s need for N and K in its metabolism, demonstrating the high consumption of K, as it activates more than 50 enzymes that have reduced absorption levels in competition. Thus, since much of this K is in the soluble form, it is relocated from older leaves to younger ones in the growth region, attempting to supply the low amount of this nutrient. However, in plants deficient in this nutrient, chemical changes occur, such as the accumulation of soluble carbohydrates, a reduction in starch content, accumulation of nitrogenous compounds, and consequent negative effects on productivity and the quality of the harvested beans [33].

### 2.4. Competitiveness Indices

The relative competitiveness (CR), clustering coefficient (K_bean_ and K_weed_), and aggressiveness coefficient (AG) indicated a significant effect for all cultivars of the common bean in association with sourgrass for PH, LA, and DM (Table 5). The crop showed CR > 1, K_crop_ > K_weed_, and AG > 0 for all evaluated situations. The common bean cultivars BRS Esteio, IPR Uirapuru, IPR Urutau, BRS Estilo, IAC 1850, and IPR Tangará showed higher competitiveness relative to sourgrass. Similar results were found when common bean cultivars competed with *U. plantaginea* [24], *C. bonariensis* [19], and *A.* spp. [12]. According to these authors, when the common bean was in the presence of competitors, the crop plants outperformed, demonstrating higher efficiency in resource absorption from the environment and, consequently, showing greater relative growth. When sowing crops in association with weeds, with varying plant proportions, crops typically have an advantage in relative productivity, thus demonstrating that intraspecific competition exceeds interspecific competition [34,35,36]. In general, well-established agricultural crops are more competitive than weeds, as they have a uniform stand, optimal density, and emerge before the weeds [18,23,36].

The joint analysis of the data (Figure 1, Figure 2 and Figure 3 and Table 1, Table 2, Table 3, Table 4 and Table 5 demonstrates the negative effects of competition with sourgrass on common bean cultivars. In other words, the weed species exhibits high competitive ability relative to the crop. Understanding the dynamics and competitiveness between plants, particularly between the common bean and sourgrass, will assist in decision-making regarding weed control at specific densities, ensuring that they do not cause negative interference with the crop, especially since this weed species includes biotypes that are resistant to EPSPs and ACCase inhibitors in various Brazilian regions where beans and other agricultural crops are grown [14].

It is also noteworthy that, in a community, there are benefits in resource competition for plants that establish themselves first, as certain intrinsic characteristics of each cultivar or hybrid, regarding its competitive ability, may lead to better resource utilization by a particular species within an ecological niche [17], consequently allowing it to outcompete another species.

The results obtained for the black bean cultivars (BRS Esteio, IPR Uirapuru, and IPR Urutau) and the carioca bean cultivars (BRS Estilo, IAC 1850, and IPR Tangará) in the presence of sourgrass did not show any differentiation between cultivar groups.

## 3. Materials and Methods

### 3.1. Edaphoclimatic Traits and Experimental Design

A total of 13 experiments were conducted in a greenhouse and at the Laboratory of Sustainable Agricultural Systems of the Federal University of the Southern Border (UFFS) Campus in the city of Erechim, Brazil; the experiments were carried out during the 2020/21 agricultural season. The experimental units consisted of plastic pots with a capacity of 8 dm^3^, filled with soil collected from an agricultural area, classified as typical Red Latosol with aluminum and iron dominance [37]. The chemical and physical properties of the soil were as follows: pH in water = 4.8; organic matter (OM) = 3.5%; P = 4.0 mg dm^−3^; K = 117.0 mg dm^−3^; Al^3+^ = 0.6 cmolc dm^−3^; Ca^2+^ = 4.7 cmolc dm^−3^; Mg^2+^ = 1.8 cmolc dm^−3^; CEC (effective) = 7.4 cmolc dm^−3^; CEC (at pH 7.0) = 16.5 cmolc dm^−3^; H^+^Al = 9.7 cmolc dm^−3^; base saturation (SB) = 6.8 cmolc dm^−3^; base saturation percentage (V) = 41%; clay = 60%, sand = 15%, and silt = 25%. Soil fertilization and pH correction were carried out based on the physicochemical analysis and following technical recommendations for common bean cultivation [38]. The experimental design used was a randomized block design with four replications. The experiments were irrigated automatically, with the timing adjusted to maintain soil moisture at field capacity.

### 3.2. Species Used in Experiments, Preliminary Tests, and Assessed Variables

The tested competitors included black bean cultivars (BRS Esteio, IPR Urutau, and IPR Uirapuru) and carioca-type cultivars (BRS Estilo, IPR Tangará, and IAC 1850), all of which competed with the weed sourgrass (*D. insularis*). All bean cultivars were inoculated with *Rhizobium tropici* prior to seeding.

As a preliminary step, six experiments were conducted with the bean cultivars and one with sourgrass, all in monoculture, totaling seven experiments aimed at estimating the plant density at which shoot dry mass (DM) would reach a constant yield [17]. For this purpose, plant densities of 1, 2, 4, 8, 16, 24, 32, 40, 48, 56, and 64 plants per pot were used (equivalent to 25, 49, 98, 196, 392, 587, 784, 980, 1,176, 1,372, and 1,568 plants per m^2^, respectively). Based on the average dry mass (DM) values of the species, a constant yield was obtained at a density of 20 plants per pot for all bean cultivars and the weed, which corresponded to 463 plants m^−2^.

### 3.3. Definitive Experiments Installed in a Substitutive Series

Six additional experiments were set up to evaluate the competitive ability of black and carioca-type bean cultivars (Table 6) when grown in competition with sourgrass plants. These experiments were conducted using a replacement series design, with different combinations of cultivars and competitors, varying the relative proportions of plants per pot—100:0, 75:25, 50:50, 25:75, and 0:100, which correspond to 20:0, 15:5, 10:10, 5:15, and 0:20 plants per pot of each species, respectively—while maintaining a constant total plant density (20 plants per pot). To establish the desired densities for each treatment and ensure uniformity of the seedlings, seeds of the common beans and sourgrass were initially sown in trays and then transplanted into pots according to the proposed treatment.

Fifty days after emergence (DAE), physiological variables of the bean cultivars were measured, such as sub-stomatal CO_2_ concentration (Ci—µmol mol^−1^), stomatal conductance (Gs—mol m^−2^ s^−1^), transpiration (E—mol H_2_O m^−2^ s^−1^), and photosynthetic (A—µmol m^−2^ s^−1^) rates. Carboxylation efficiency (CE—mol CO_2_ m^−2^ s^−1^) and water-use efficiency (WUE—mol CO_2_ mol H_2_O^−1^) were calculated from the ratios of the variables A/Ci and A/E, respectively. These variables were measured in the middle third of the bean plants, on the younger fully expanded leaves. A gas analyzer (IRGA), LCA PRO (Analytical Development Co. Ltd., Hoddesdon, UK), was used to determine the physiological variables. The chlorophyll index (CI) was determined using a portable chlorophyll meter (SPAD 502 Plus, Minolta, Beijing, China) at five points on each plant in the lower, middle, and upper parts of the leaf canopy. All physiological variables were assessed in each block on a single day, between 7 and 10 AM, to ensure that the environmental conditions remained consistent during the measurements.

The plant height (PH) and leaf area (LA) of the species were also measured 50 DAE. Plant height (cm) was determined using a graduated ruler, from the soil surface to the last fully expanded leaf. For the determination of LA, an electronic leaf area meter (LICOR-3100C) was used, and the variable was quantified for all plants in each treatment. After determining LA, plants were placed in kraft paper bags and dried in a forced air circulation oven at 65 °C for 48 h, until the material reached a constant weight to measure the dry mass (DM) of the species.

For nutrient analysis—nitrogen (N), phosphorus (P), potassium (K), and carbon (C)—the previously dried leaf samples were ground in a Star FT-50 knife mill, using a sieve with a mesh diameter of 0.5 mm. Leaf nutrients of the common beans were determined according to the methodologies proposed by Tedesco et al. [39].

### 3.4. Experimental Analysis

The data were analyzed using the graphical analysis method of variation or relative productivity [17,23,40]. This procedure, also known as the conventional method for replacement series experiments, involves constructing a diagram based on relative productivity (PR) and total relative productivity (PRT). When the PR result is a straight line, it indicates that the species’ abilities are equivalent. If the PR results in a concave line, it indicates a growth disadvantage for one or both species. Conversely, if the PR shows a convex line, there is a growth benefit for one or both species. When the PRT is equal to 1 (straight line), competition occurs for the same resources; if it is greater than 1 (convex line), competition is avoided. If the PRT is less than 1 (concave line), mutual growth disadvantage occurs [17,23,40].

The relative competitiveness index (CR), relative clustering coefficient (K), and aggressiveness (A) of the species were also calculated. CR represents the comparative growth of the common bean cultivar (X) in relation to the competitor sourgrass (Y); K indicates the relative dominance of one species over the other; and A identifies which species is more aggressive. Thus, the indices CR, K, and A show which species is more competitive, and their joint interpretation provides a more reliable assessment of the species’ competitiveness [17,40]. The common bean cultivars (X) are more competitive than the sourgrass competitor (Y) when CR > 1, K_cultivar_ > K_weed_, and A > 0; on the other hand, the competitor (Y) is more competitive than the common bean cultivars (X) when CR < 1, K_cultivar_ < K_weed_, and A < 0 [17,18]. To calculate these indices, 50:50 proportions of the species involved in the experiment (common bean versus sourgrass) were used, i.e., densities of 10:10 plants per pot, applying the following equations: CR = PRx/Pry; Kx = PRx/(1 − PRx); Ky = Pry/(1 − Pry); A = PRx − Pry [17,41]. 

The statistical analysis procedure for relative productivity or variation included the calculation of the differences for the PR values (DPR) obtained at the 25%, 50%, and 75% proportions in relation to the values on the hypothetical line at the respective proportions, which were 0.25, 0.50, and 0.75 for PR [17,18]. The “*t*” test was used to test the differences related to the indices DPR, PRT, CR, K, and A [17]. The null hypothesis considered for testing the differences in DPR and A was that the means were equal to zero (Ho = 0); for PRT and CR, that the means were equal to one (Ho = 1); and for K, that the means of the differences between K_cultivar_ and K_weed_ were equal to zero [Ho = (K_cultivar_ − K_weed_) = 0]. The criterion for considering the PR and PRT curves as different from the hypothetical lines was that, in at least two proportions, there were significant differences according to the “*t*” test [17,18]. Similarly, for the indices CR, K, and A, the existence of differences in competitiveness was considered when, in at least two of them, there was a significant difference according to the “*t*” test.

The results obtained for the variables—morphological (PH, LA, and DM), physiological (CI, Ci, Gs, E, A, CE, and WUE), and nutrient contents (N, P, K, and C)—expressed as mean values per treatment, were subjected to analysis of variance using the F-test. If significant, means were compared using Dunnett’s test, considering the monocultures as controls in these comparisons. A significance level of *p* ≤ 0.05 was adopted for all statistical analyses.

## 4. Conclusions

There is equivalence in the competition mechanisms and demand for environmental resources among common bean cultivars when infested with sourgrass.

The physiological variables of bean plants are negatively affected by increasing sourgrass density in competition with the crop.

The common bean exhibits greater relative growth than sourgrass, as indicated by relative competitiveness indices (CR, K, and A).

Nitrogen, phosphorus, potassium, and carbon levels are influenced by both the bean cultivar and the proportion of the competing species.

Overall, higher sourgrass density negatively impacts gas exchange in bean cultivars.

Interspecific competition causes more severe damage to morphological, physiological, and nutritional traits than intraspecific competition.

Further field studies are needed to determine the most effective growth stage for sourgrass control, and to assess the economic impact of high weed densities on the same bean cultivars evaluated in this study. These findings will provide valuable guidance to producers on the ideal timing and weed density thresholds for effective sourgrass management, emphasizing the importance of timely control to minimize economic losses and maintain product quality.

## Figures and Tables

**Figure 1 plants-14-02684-f001:**
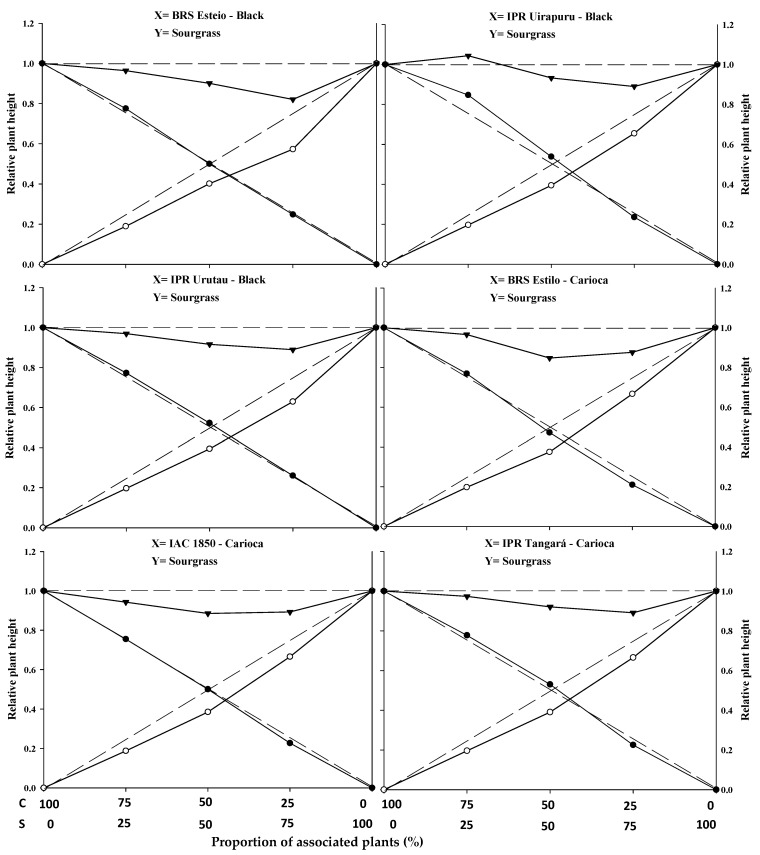
Relative productivity (RP) for relative plant height of common beans (●) and sourgrass—*Digitaria insularis* (○), and total relative productivity (TRP) of the community (►) as a function of plant proportion (common bean–sourgrass). Dashed lines represent the expected values in the absence of competition, and solid lines represent the observed values when the species competed at different plant proportions. C: common bean; S: sourgrass.

**Figure 2 plants-14-02684-f002:**
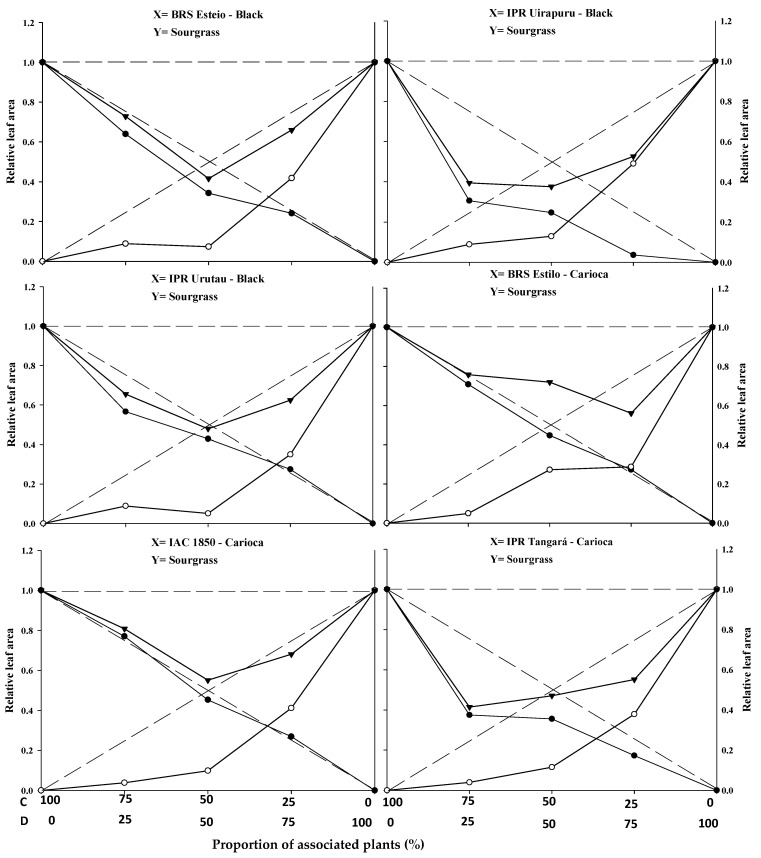
Relative productivity (RP) for relative leaf area of black and carioca bean plants (●) and sourgrass—*Digitaria insularis* (○), and total relative productivity (TRP) of the community (►) as a function of plant proportion (common bean–sourgrass). Dashed lines represent the expected values in the absence of competition, and solid lines represent the observed values when the species competed at different plant proportions. C: common bean; D: sourgrass.

**Figure 3 plants-14-02684-f003:**
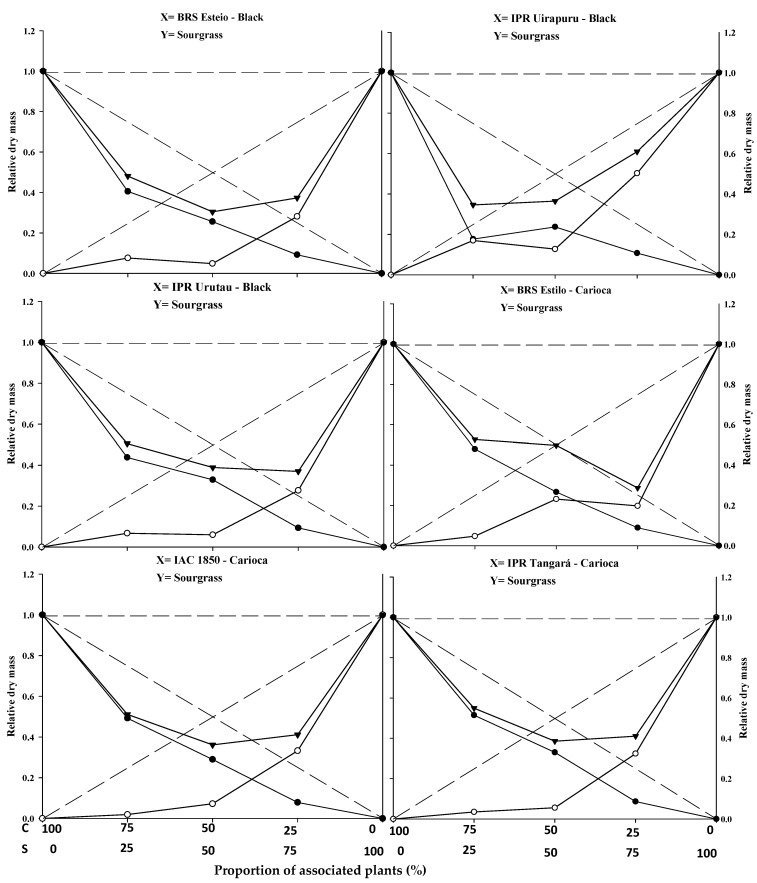
Relative productivity (RP) for relative dry mass of black and carioca bean plants (●) and sourgrass—*Digitaria insularis* (○), and total relative productivity (TRP) of the community (►) based on the plant proportion (common bean–sourgrass). Dashed lines represent the expected values in the absence of competition, and solid lines represent the observed values when the species competed at different plant proportions. C: common bean; S: sourgrass.

**Table 1 plants-14-02684-t001:** Relative differences for the variables height, leaf area, and shoot dry mass of black bean cultivars (BRS Esteio, IPR Uirapuru, and IPR Urutau), carioca bean cultivars (BRS Estilo, IAC 1850, and IPR Tangará), and the competitor sourgrass (*Digitaria insularis*), 50 days after crop emergence. UFFS, Erechim Campus, RS.

Variable	Plant Proportion (Bean/Sourgrass)		
	75:25	50:50	25:75
Plant Height (PH—cm)			
BRS Esteio	0.05 (±0.01) *	0.03 (±0.001) *	0.04 (±0.001) *
Sourgrass	−0.06 (±0.001) *	−0.13 (±0.02) *	−0.08 (±0.01) *
Total	0.96 (±0.01) *	0.90 (±0.01) *	0.82 (±0.001) *
IPR Uirapuru	0.10 (±0.01)*	0.04 (±0.01)*	−0.01 (±0.001) *
Sourgrass	−0.05 (±0.001) *	−0.11 (±0.001) *	−0.10 (±0.001) *
Total	1.04 (±0.01) *	0.93 (±0.001) *	0.89 (±0.001) *
IPR Urutau	0.05 (±0.01) *	0.04 (±0.001) *	0.02 (±0.01)
Sourgrass	−0.06 (±0.001) *	−0.10 (±0.01) *	−0.11 (±0.02) *
Total	0.97 (±0.03)	0.92 (±0.01) *	0.89 (±0.001) *
BRS Estilo	0.05 (±0.02)	0.06 (±0.02)	0.02 (±0.01)
Sourgrass	−0.06 (±0.01) *	0.06 (±0.02)	0.02 (±0.01)
Total	0.97 (±0.001) *	0.85 (±0.001) *	0.88 (±0.001) *
IPR IAC 1850	0.06 (±0.01) *	0.002 (±0.04)	−0.002 (±0.01)
Sourgrass	−0.08 (±0.02) *	−0.08 (±0.01) *	−0.14 (±0.03) *
Total	0.94 (±0.001) *	0.89 (±0.001) *	0.89 (±0.001) *
IPR Tangará	0.02 (±0.01)	0.02 (±0.01)	0.01 (±0.001) *
Sourgrass	−0.04 (±0.01) *	−0.08 (±0.01) *	−0.08 (±0.01) *
Total	0.97 (±0.01)	0.92 (±0.001) *	0.89 (±0.001) *
Leaf Area (LA—cm^2^ per pot)			
BRS Esteio	0.05 (±0.01) *	0.03 (±0.001) *	0.04 (±0.001) *
Sourgrass	−0.06 (±0.001) *	−0.13 (±0.02) *	−0.08 (±0.01) *
Total	0.73 (±0.01) *	0.42 (±0.001) *	0.66 (±0.03) *
IPR Uirapuru	−0.02 (±0.01)	0.04 (±0.02) *	0.01 (±0.001) *
Sourgrass	−0.07 (±0.01) *	−0.08 (±0.001) *	−0.09 (±0.01) *
Total	0.39 (±0.01) *	0.38 (±0.001) *	0.53 (±0.001) *
IPR Urutau	0.05 (±0.01) *	0.04 (±0.001) *	0.02 (±0.01)
Sourgrass	−0.06 (±0.001) *	−0.10 (±0.01) *	−0.11 (±0.02) *
Total	0.65 (±0.01) *	0.48 (±0.001) *	0.62 (±0.01) *
BRS Estilo	0.05 (±0.02)	0.06 (±0.02)	0.02 (±0.01)
Sourgrass	−0.06 (±0.01) *	−0.08 (±0.001) *	−0.08 (±0.01) *
Total	0.76 (±0.001) *	0.72 (±0.02) *	0.56 (±0.01) *
IPR IAC 1850	0.02 (±0.001) *	−0.05 (±0.01) *	0.02 (±0.001) *
Sourgrass	−0.21 (±0.001) *	−0.40 (±0.001) *	−0.34 (±0.02) *
Total	0.81 (±0.001) *	0.55 (±0.001) *	0.68 (±0.03) *
IPR Tangará	0.02 (±0.01) *	0.02 (±0.01) *	0.01 (±0.001) *
Sourgrass	−0.04 (±0.01) *	−0.08 (±0.01) *	−0.08 (±0.01) *
Total	0.41 (±0.01) *	0.47 (±0.01) *	0.55 (±0.02) *
Shoot Dry Mass (DM—g per pot)			
BRS Esteio	−0.29 (±0.001) *	−0.22 (±0.01) *	−0.13 (±0.001) *
Sourgrass	−0.23 (±0.001) *	−0.37 (±0.001) *	−0.48 (±0.001) *
Total	0.48 (±0.01) *	0.30 (±0.01) *	0.37 (±0.02) *
IPR Uirapuru	−0.23 (±0.01) *	−0.20 (±0.001) *	−0.10 (±0.001) *
Sourgrass	−0.24 (±0.001) *	−0.44 (±0.001) *	−0.57 (±0.01) *
Total	0.35 (±0.03) *	0.36 (±0.001) *	0.61 (±0.02) *
IPR Urutau	−0.27 (±0.01) *	−0.18 (±0.01) *	−0.11 (±0.001) *
Sourgrass	−0.23 (±0.001) *	−0.43 (±0.001) *	−0.55 (±0.02) *
Total	0.51 (±0.02) *	0.39 (±0.02) *	0.37 (±0.02) *
BRS Estilo	−0.03 (±0.02)	−0.17 (±0.001) *	−0.06 (±0.01) *
Sourgrass	−0.24 (±0.001) *	−0.43 (±0.001) *	−0.50 (±0.001) *
Total	0.53 (±0.001) *	0.50 (±0.01) *	0.29 (±0.01) *
IPR IAC 1850	−0.06 (±0.02) *	−0.18 (±0.01) *	−0.07 (±0.01) *
Sourgrass	−0.24 (±0.001) *	−0.41 (±0.01) *	−0.56 (±0.01) *
Total	0.51 (±0.02) *	0.36 (±0.01) *	0.41 (±0.01) *
IPR Tangará	−0.15 (±0.01) *	−0.19 (±0.001) *	−0.11 (±0.001) *
Sourgrass	−0.23 (±0.001) *	−0.42 (±0.001) *	−0.55 (±0.03) *
Total	0.55 (±0.02) *	0.39 (±0.01) *	0.41 (±0.01) *

* Significant difference between the expected (theoretical) and the observed (experimental) data by the “*t*” test (*p* ≤ 0.05). Values in brackets represent the standard error of the mean. In Figure 1, Figure 2 and Figure 3, the expected and the observed values are represented by dashed (- - - -) and solid (―――) lines, respectively. Differences between the crop and the weed are considered significant when there is difference between at least two of the three means in the line (75:25, 50:50, 25:75), as indicated by blank cell backgrounds. Non-significant differences between crop and weed are marked as grey background cells.

**Table 2 plants-14-02684-t002:** Differences between associated or non-associated plants of black bean cultivars (BRS Esteio, IPR Uirapuru, and IPR Urutau), carioca bean cultivars (BRS Estilo, IAC 1850, and IPR Tangará), and sourgrass (*Digitaria insularis*) for the variables height, leaf area, and shoot dry mass of the plants. UFFS, Erechim Campus, RS.

Plant Proportion Bean/Sourgrass	Plant Height (cm)		Leaf Area (cm^2^ pot^−1^)		Dry Mass (g pot^−1^)	
	BRS Esteio	Sourgrass	BRS Esteio	Sourgrass	BRS Esteio	Sourgrass
100:0 (T)	58.80	89.47	7033.43	9541.46	104.75	55.98
75:25	60.80 *	68.33 *	5999.87 *	5321.38 *	56.57 *	21.00 *
50:50	58.80	71.80 *	4819.42 *	1399.95 *	53.55 *	5.43 *
25:75	58.33	67.60 *	6793.78	3384.18 *	38.57 *	17.00 *
CV (%)	1.66	1.49	7.60	14.34	3.31	17.59
	IPR Uirapuru	Sourgrass	IPR Uirapuru	Sourgrass	IPR Uirapuru	Sourgrass
100:0 (T)	53.13	89.47	13711.96	9541.46	69.30	55.98
75:25	60.00 *	78.10 *	5591.60 *	6237.56 *	16.30 *	37.50 *
50:50	57.20 *	70.60 *	6768.25 *	2462.51 *	32.87 *	14.25 *
25:75	50.00 *	70.40 *	1985.79 *	3384.18 *	29.78 *	38.00 *
CV (%)	2.64	0.50	4.32	11.67	6.70	9.37
	IPR Urutau	Sourgrass	IPR Urutau	Sourgrass	IPR Urutau	Sourgrass
100:0 (T)	54.87	89.47	7947.31	9541.46	106.05	55.98
75:25	56.53	75.12 *	5999.87 *	4451.85 *	61.95 *	20.63 *
50:50	57.33	70.53 *	6815.58 *	963.13 *	69.78 *	6.70 *
25:75	57.07	70.50 *	8734.09 *	3384.18 *	39.53 *	15.07 *
CV (%)	4.24	1.16	2.85	14.63	7.07	14.49
	BRS Estilo	Sourgrass	BRS Estilo	Sourgrass	BRS Estilo	Sourgrass
100:0 (T)	57.20	89.47	5658.11	9541.46	100.58	55.98
75:25	58.60 *	79.50 *	5341.95	3655.29 *	64.27 *	14.75 *
50:50	54.00 *	67.10 *	5059.33	5200.43 *	53.60 *	25.90 *
25:75	47.98 *	70.53 *	6177.65	1892.07 *	35.87 *	10.57 *
CV (%)	0.82	0.59	8.47	14.87	17.19	11.89
	IAC 1850	Sourgrass	IAC 1850	Sourgrass	IAC 1850	Sourgrass
100:0 (T)	57.80	89.47	6702.65	9541.46	95.15	55.98
75:25	58.20	79.40 *	6880.73	5238.19	62.45 *	24.83 *
50:50	57.87	68.90 *	6068.31	1873.47 *	55.10 *	8.03 *
25:75	52.53 *	67.13 *	7206.25	1460.47 *	29.90 *	4.23 *
CV (%)	1.08	0.67	9.07	14.37	5.78	15.28
	IPR Tangará	Sourgrass	IPR Tangará	Sourgrass	IPR Tangará	Sourgrass
100:0 (T)	54.40	89.47	8753.96	9541.46	95.37	55.98
75:25	56.40 *	79.40 *	4377.21 *	4819.41 *	65.43 *	24.20 *
50:50	57.68 *	69.90 *	6221.16 *	2198.89 *	63.03 *	6.23 *
25:75	49.00 *	70.13 *	6045.55 *	1495.31 *	33.05 *	7.80 *
CV (%)	1.77	1.32	7.87	15.42	6.14	15.28

* Means differ from the control (T) by Dunnett’s test (*p* ≤ 0.05).

**Table 3 plants-14-02684-t003:** Differences between associated or non-associated bean cultivars in competition with sourgrass (*Digitaria insularis*) for plant physiology-related variables: chlorophyll content (SPAD), sub-stomatal CO_2_ concentration (Ci—µmol mol^−1^), photosynthetic rate (A—µmol m^−2^ s^−1^), stomatal conductance (Gs—mol m^−2^ s^−1^), transpiration rate (E—mol H_2_O m^−2^ s^−1^), carboxylation efficiency (CE—mol CO_2_ m^−2^ s^−1^), and water-use efficiency (WUE—mol CO_2_ mol H_2_O^−1^). UFFS, Erechim Campus, RS.

Plant Proportion Bean/Sourgrass	Physiological Parameter						
	SPAD	Ci	Gs	E	A	CE	WUE
	BRS Esteio—Black Type						
100:0 (T)	35.75	332.67	0.79	2.60	13.25	0.04	5.10
75:25	40.27 *	290.67 *	0.53 *	4.62 *	15.36 *	0.05 *	3.33 *
50:50	34.07	256.33 *	0.37 *	4.80 *	18.10 *	0.07 *	3.77 *
25:75	39.80 *	250.00 *	0.33 *	4.55 *	17.68 *	0.07 *	3.89 *
CV (%)	2.41	1.86	2.91	2.63	13.15	4.88	3.34
	IPR Uirapuru—Black Type						
100:0 (T)	34.43	293.33	0.56	3.73	16.22	0.06	4.35
75:25	36.47	296.33	0.41 *	4.44 *	13.53 *	0.05	3.05 *
50:50	35.30	272.00 *	0.37 *	4.80 *	15.91	0.06	3.31 *
25:75	38.00 *	268.50 *	0.29 *	4.28 *	15.73	0.06	3.69 *
CV (%)	4.33	2.91	3.41	2.63	8.21	9.69	9.20
	IPR Urutau—Black Type						
100:0 (T)	36.70	212.00	0.66	3.26	14.99	0.05	4.61
75:25	35.33	293.00 *	0.46 *	4.62 *	15.86	0.05	3.43 *
50:50	34.23 *	259.50 *	0.33 *	4.65 *	15.87	0.06 *	3.41 *
25:75	41.30 *	282.00 *	0.31 *	4.45 *	12.60 *	0.04	2.83 *
CV (%)	2.45	1.90	5.80	1.97	8.09	8.99	10.88
	BRS Estilo—Carioca Type						
100:0 (T)	35.35	308.50	0.52	3.78	13.05	0.04	3.45
75:25	33.90 *	278.33 *	0.41 *	4.57 *	15.32 *	0.06 *	3.35
50:50	34.00 *	272.00 *	0.36 *	4.55 *	15.84 *	0.06 *	3.48
25:75	38.85 *	260.00 *	0.43 *	5.01 *	19.48 *	0.07 *	3.89 *
CV (%)	0.83	0.59	1.61	1.68	3.34	3.32	3.17
	IPR Tangará—Carioca Type						
100:0 (T)	35.87	294.33	0.51	3.96	15.75	0.05	3.98
75:25	38.10 *	276.50 *	0.46 *	5.04 *	16.84 *	0.06 *	3.34 *
50:50	39.40 *	276.00 *	0.42 *	4.77 *	16.46 *	0.06 *	3.45 *
25:75	42.80 *	260.67 *	0.41 *	5.12 *	18.75 *	0.07 *	3.67 *
CV (%)	1.82	1.22	4.06	2.04	1.57	2.20	2.07
	IAC 1850—Carioca Type						
100:0 (T)	35.00	293.00	0.50	4.26	15.05	0.05	3.54
75:25	34.23	292.50	0.44 *	5.12 *	12.63 *	0.04	2.47 *
50:50	36.80	267.33 *	0.27 *	4.03	15.13	0.05	3.77
25:75	36.60	267.50 *	0.38 *	5.41 *	16.22	0.06 *	3.00 *
CV (%)	2.76	3.02	7.83	4.75	7.91	9.05	8.70

* Means differ from the control (T) by Dunnett’s test (*p* ≤ 0.05).

**Table 4 plants-14-02684-t004:** Responses of bean cultivars to interference from sourgrass (*Digitaria insularis*), expressed by nutrient levels (nitrogen—N, phosphorus—P, potassium—K and carbon—C) in plants grown in replacement series experiments. UFFS, Erechim Campus/RS.

Plant Proportion Bean/Sourgrass	Nutrient Content (dag kg^−1^—%)			
	N	P	K	C
	BRS Esteio—Black Type			
100:0 (T)	3.23	0.37	0.83	2.04
75:25	3.74	0.43	1.11 *	1.80
50:50	4.34 *	0.59 *	1.22 *	1.85
25:75	4.74 *	0.52 *	1.44 *	1.70 *
CV (%)	8.30	14.71	9.40	2.33
	IPR Uirapuru—Black Type			
100:0 (T)	6.50	0.61	1.42	1.80
75:25	4.86 *	0.67	1.18 *	1.83
50:50	3.23 *	0.37 *	1.10 *	1.85
25:75	3.74 *	0.43 *	1.16 *	1.96 *
CV (%)	8.13	16.15	9.98	3.41
	IPR Urutau—Black Type			
100:0 (T)	4.64	0.63	1.04	1.82
75:25	4.20	0.56	1.12	1.85
50:50	4.42	0.49	0.99	1.84
25:75	4.60	0.51	1.24	1.81
CV (%)	10.74	14.21	11.96	1.48
	BRS Estilo—Carioca Type			
100:0 (T)	4.34	0.59	1.10	1.96
75:25	4.74	0.52	1.04	1.94
50:50	4.64	0.63	1.10	2.03 *
25:75	4.20	0.59	1.02	2.01
CV (%)	8.42	14.52	10.32	1.63
	IPR Tangará—Carioca Type			
100:0 (T)	4.42	0.49	0.92	1.92
75:25	4.60	0.51	1.03	1.93
50:50	6.50 *	0.61	1.02	1.92
25:75	4.86	0.67 *	0.96	1.77 *
CV (%)	9.46	16.47	9.34	1.72
	IAC 1850—Carioca Type			
100:0 (T)	4.12	0.59	1.21	1.81
75:25	5.74 *	0.67	1.15	1.78
50:50	4.83	0.65	1.23	1.75
25:75	4.40	0.65	1.02 *	1.73
CV (%)	9.57	10.21	8.25	3.13

* Means differ from the control (T) by Dunnett’s test (*p* ≤ 0.05).

**Table 5 plants-14-02684-t005:** Competitiveness indices between black bean cultivars (BRS Esteio, IPR Uirapuru, and IPR Urutau) and carioca bean cultivars (BRS Estilo, IAC 1850, and IPR Tangará) and the competitor sourgrass (*Digitaria insularis*), expressed by relative competitiveness (CR), relative clustering coefficients (K), and aggressiveness (AG), obtained in experiments conducted in replacement series. UFFS, Erechim Campus/RS.

Plant Species	CR	K_x_ (Bean)	K_y_ (Sourgrass)	AG
Plant Height (PH)				
BRS Esteio × sourgrass	1.25 (±0.02) *	1.00 (±0.001) *	0.67 (±0.02) *	0.10 (±0.01) *
IPR Uirapuru × sourgrass	1.36 (±0.001) *	1.17 (±0.04) *	0.65 (±0.002) *	0.14 (±0.001) *
IPR Urutau × sourgrass	1.33 (±0.01) *	1.09 (±0.01) *	0.65 (±0.01) *	0.13 (±0.001) *
BRS Estilo × sourgrass	1.26 (±0.001) *	0.89 (±0.001) *	0.60 (±0.003) *	0.10 (±0.001) *
IAC 1850 × sourgrass	1.30 (±0.01) *	1.00 (±0.003) *	0.63 (±0.003) *	0.12 (±0.001) *
IPR Tangará × sourgrass	1.36 (±0.001) *	1.13 (±0.05) *	0.64 (±0.003) *	0.14 (±0.001) *
Leaf Area (LA)				
BRS Esteio × sourgrass	4.70 (±0.24) *	0.52 (±0.004) *	0.08 (±0.004) *	0.27 (±0.001) *
IPR Uirapuru × sourgrass	1.91 (±0.04) *	0.33 (±0.006) *	0.15 (±0.001) *	0.12 (±0.001) *
IPR Urutau × sourgrass	8.74 (±0.86) *	0.75 (±0.01) *	0.05 (±0.006) *	0.38 (±0.01) *
BRS Estilo × sourgrass	1.69 (±0.17) *	0.81 (±0.001) *	0.38 (±0.04) *	0.17 (±0.02) *
IAC 1850 × sourgrass	4.62 (±0.15) *	0.83 (±0.02) *	0.11 (±0.002) *	0.35 (±0.01) *
IPR Tangará × sourgrass	3.08 (±0.07) *	0.55 (±0.02) *	0.13 (±0.002) *	0.24 (±0.01) *
Dry Mass (DM)				
BRS Esteio × sourgrass	5.30 (±0.32) *	0.34 (±0.01) *	0.05 (±0.002) *	0.21 (±0.01) *
IPR Uirapuru × sourgrass	1.86 (±0.03) *	0.31 (±0.002) *	0.15 (±0.003) *	0.11 (±0.001) *
IPR Urutau × sourgrass	5.50 (±0.20) *	0.49 (±0.03) *	0.06 (±0.002) *	0.27 (±0.01) *
BRS Estilo × sourgrass	1.16 (±0.03) *	0.36 (±0.001) *	0.30 (±0.01) *	0.04 (±0.01) *
IAC 1850 × sourgrass	4.16 (±0.47) *	0.41 (±0.02) *	0.08 (±0.008) *	0.22 (±0.01) *
IPR Tangará × sourgrass	5.97 (±0.35) *	0.50 (±0.03) *	0.06 (±0.003) *	0.27 (±0.01) *

For AG: * = significant difference from “0” by the *t*-test at 5% probability. For Kx/Ky: * = significant difference between the respective Kx and Ky means by the *t*-test at 5% probability. For CR: * = significant difference from “1” by the *t*-test at 5% probability.

**Table 6 plants-14-02684-t006:** Genetic characteristics of the common bean cultivars used in the study. UFFS, Erechim Campus/RS.

Company/Owner	Pedigree	Group	Cycle	Growth Habit	Type
Embrapa	BRS Esteio	Preto	Normal	Indeterminate	II
IAPAR	IPR Urutau	Preto	Medium	Indeterminate	II
IAPAR	IPR Uirapuru	Preto	Medium	Indeterminate	II
Embrapa	BRS Estilo	Carioca	Medium	Indeterminate	II
IAPAR	IPR Tangará	Carioca	Medium	Indeterminate	II
Instituto Agronômico	IAC 1850	Carioca	Medium	Indeterminate	II

## Data Availability

The data presented in this study are available from the corresponding author upon reasonable request.

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
