# Peer review of "Morphophysiological and Nutritional Responses of Bean Cultivars in Competition with Digitaria insularis"

_plants, 2025, doi:10.3390/plants14172684_

Round 1
Reviewer 1 Report
Comments and Suggestions for Authors
- Title to be changed - regarding the format, why all capital letters?
- Graphical summary representing the effect of D. insularis competition on morphophysiological and nutritional characteristics of common bean - why at the very beginning? Is it included in the abstract? There is no author information?
- Why is page 3 so blank? Please start the methods on page 3, continuing the manuscript, and not leaving such a gap.
- Single-year studies? How many seasons, 2020 and 2021? Or at the turn of the year?
- Can we already speak of concrete conclusions after a one-year study? In my opinion, this is preliminary research, and it's worth mentioning that.
- References - please submit and file according to the journal's requirements.
- How was soil moisture maintained in pots?
- Were the seeds treated with Rhizobium?
- 13 experiments? What were the differences between these experiments? Because I have the impression that variants of certain experiments are being described.
-….Six additional experiments were set up to evaluate the competitive ability of black and carioca-type bean cultivars (Table 1) when grown in competition with sourgrass plants. – How did these experiments differ? Are we talking about variants? One experience with different variants? Second experience with different variants?
-….. As a preliminary step, six experiments were conducted with the bean cultivars and one with sourgrass, all in monoculture, totaling seven experiments aimed at estimating the plant density at which shoot dry mass (DM) would achieve a constant yield.
- I can't find information about the 13th experiment?
10. Difficult-to-analyze graphs when the X-axis is only described in the last row of the graphs (Figure 1, Figure 2, Figure 3)
11. Was there a specific soil moisture content in the pots?
12. Supplementary file(s)- Raw results document??
Author Response
Dear Editor and Reviewers,
We are resubmitting the carefully revised manuscript. Almost all suggestions have been accepted and incorporated into the text. All changes requested by the editor and reviewers are highlighted using the track changes feature.We would like to thank the reviewers for their valuable contributions, which greatly helped us improve the quality of our manuscript.
Sincerely,
Leandro Galon
Below and attached are the responses from Reviewer I:
- Title to be changed - regarding the format, why all capital letters? Action Taken: Corrected, according to reviewer’s request;
- Graphical summary representing the effect of D. insularis competition on morphophysiological and nutritional characteristics of common bean - why at the very beginning? Is it included in the abstract? There is no author information? Action Taken: We removed the graphical abstract from the manuscript and uploaded it separately. We also included the authorship credit, which corresponds to the same authors of the manuscript, Gismael Francisco Perin and Leandro Galon.
- Why is page 3 so blank? Please start the methods on page 3, continuing the manuscript, and not leaving such a gap.Action Taken: Corrected, according to reviewer’s request.
- Single-year studies? How many seasons, 2020 and 2021? Or at the turn of the year? Action Taken: The experiments were conducted from October 2020 to February 2021. It was rewritten in the text.
- Can we already speak of concrete conclusions after a one-year study? In my opinion, this is preliminary research, and it's worth mentioning that. Action Taken: The authors have extensive experience researching competition between cultivated plants and weeds, forming one of the most productive teams in Brazil in this field. The lead researchers have completed master's and doctoral degrees focused on the competitive ability of various agricultural crops and continue publishing in recognized scientific journals. Over the years, we have observed that, due to the controlled environment and low variability, there is no need to repeat such experiments if they are consistent / have a low coefficient of variation (as is the case in the present study). Moreover, this study includes 13 experiments, not just one. However, field replication is recommended, but it is beyond the scope of the present work.
- References - please submit and file according to the journal's requirements. Action Taken: Corrected, according to reviewer’s request.
- How was soil moisture maintained in pots? Action Taken: The experiments were irrigated automatically, with the timing adjusted to maintain soil moisture at field capacity. This information was added to the manuscript.
- Were the seeds treated with Rhizobium? Action Taken: Yes, all cultivars and their seeds were inoculated with Rhizobium tropici. This information has been added to the manuscript.
- 13 experiments? What were the differences between these experiments? Because I have the impression that variants of certain experiments are being described. ..Six additional experiments were set up to evaluate the competitive ability of black and carioca-type bean cultivars (Table 1) when grown in competition with sourgrass plants. – How did these experiments differ? Are we talking about variants? One experience with different variants? Second experience with different variants? ... As a preliminary step, six experiments were conducted with the bean cultivars and one with sourgrass, all in monoculture, totaling seven experiments aimed at estimating the plant density at which shoot dry mass (DM) would achieve a constant yield. - I can't find information about the 13th experiment? Action Taken: The experiments were as follows: Preliminary experiments: Bean cultivars: 6 and Weed (Sourgrass): 1 = 7; Main experiments: Bean cultivars x Weed: 6; Total: 13. We have highlighted in red the experiments in your question, for reference.
- Difficult-to-analyze graphs when the X-axis is only described in the last row of the graphs (Figure 1, Figure 2, Figure 3). Action Taken: We thought it best to include only the X-axis values in the last graphs because there are many numbers and data as legends, which would cause significant clutter in the other figures. Additionally, including all would make the figure too large to fit on a single page. Furthermore, the proportions indicated in the X-axis legend are marked in each graph by •, â—¦ and â–¼.
- Was there a specific soil moisture content in the pots? Action Taken: Soil moisture in the pots was maintained at about 80% of field capacity.
- Supplementary file(s)- Raw results document? Action Taken: During the submission process, we uploaded the raw results as supplementary documents, since all analyzed results are included in the article.

Reviewer 2 Report
Comments and Suggestions for Authors
In the manuscript “MORPHYSIOLOGICAL AND NUTRITIONAL RESPONSES OF BEAN CULTIVARS IN COMPETITION WITH Digitaria insularis”, the authors investigated the responses of bean cultivars in competition with Digitaria insularis. The authors did a lot of work, and the results are meaningful. However, there are still some points that need to be clarified before the next step.
- Please capitalize only the first letter of the first word and “Digitaria insularis”, and change the remaining letters in the title to lowercase.
- The results in the abstract part are not clearly described. Some accurate data may be added to make the results more straightforward.
- The words in the keywords part are limited; I suggest that more key words may be supplemented.
- Some sentences are too long to get across, for example, “Lines 50-52”, “Lines 56-58”, and these sentences should be rewritten.
- The authors indicated that they determined nutrient contents, however, only some basic nutrient elements were analyzed.
- The y-axis title of the first graph in Figure 1 is incorrect; please verify.
- Table 2 is inappropriate here. Generally, differences are marked after the mean values or figures.
- The conclusion part is too long and has too many paragraphs. This part should conclude the main findings of the work.
- The discussion is not strong throughout the result and discussion part. More detailed and deeply discussion is required for taking the meaning of the work further.
Author Response
Dear Editor and Reviewers,
We are resubmitting the carefully revised manuscript. Almost all suggestions have been accepted and incorporated into the text. All changes requested by the editor and reviewers are highlighted using the track changes feature.We would like to thank the reviewers for their valuable contributions, which greatly helped us improve the quality of our manuscript.
Sincerely,
Leandro Galon
Below and attached are the responses from Reviewer II:
- Please capitalize only the first letter of the first word and “Digitaria insularis”, and change the remaining letters in the title to lowercase. Action Taken: Corrected, according to reviewer’s request.
- The results in the abstract part are not clearly described. Some accurate data may be added to make the results more straightforward. Action Taken: We have fully rewritten the Results into the Abstract, and inserted an overall mean of cultivars for some parameters.
- The words in the keywords part are limited; I suggest that more key words may be supplemented. Action Taken: We have reviewed the keywords and also included new ones.
- Some sentences are too long to get across, for example, “Lines 50-52”, “Lines 56-58”, and these sentences should be rewritten. Action Taken: We have included breaks in the texts, as requested, for easier understanding.
- The authors indicated that they determined nutrient contents, however, only some basic nutrient elements were analyzed. Action Taken: The essential nutrients and those demanded in greater quantities by plants—where high competition usually occurs—were determined, especially nitrogen, which both sourgrass and common bean intensely compete for. Based on results from other studies, no competition effects were typically observed for some due to competition. WE focused on the most significant ones for plant-weed interaction. The nutrients analyzed are described in Materials and Methods, line 154.
- The y-axis title of the first graph in Figure 1 is incorrect; please verify. Action Taken: Corrected, according to reviewer’s request.
- Table 2 is inappropriate here. Generally, differences are marked after the mean values or figures. Action Taken: Surely the reviewer is also correct here. It is usual for the authors, and we have forgotten to explain it better.
Table 2:
* Significant difference between the expected (theoretical) and the observed (experimental) data by the “t” test (p≤0.05). Values into brackets represent the standard error of the mean. In Figures 1 to 3, the expected and the observed values are represented by dashed ( - - - - ) and solid (―――) lines, respectively. Differences between the crop and the weed are considered significant when there is difference between in at least two of the three means in the line (75:25, 50:50, 25:75), as indicated by blank cell backgrounds. Non-significant differences between crop and weed are marked as grey background cells.Table 6:
For AG: * = significant difference from “0” by the t-test at 5% probability; For Kx/Ky: * = significant difference between the respective Kx and Ky means, by the t-test at 5% probability; For CR: * = significant difference from “1” by the t-test, at 5% probability.
These were added as footnotes to the respective tables. - The conclusion part is too long and has too many paragraphs. This part should conclude the main findings of the work. Action Taken: We have reviewed and reduced the conclusions.
- The discussion is not strong throughout the result and discussion part. More detailed and deeply discussion is required for taking the meaning of the work further. Action Taken: We appreciate the reviewer’s suggestion regarding the need for a more detailed and in-depth discussion. However, due to the overall length of the manuscript, which is already around 30 pages, we were required to limit the extent of the discussion and focus on the most relevant and significant findings. We have ensured that the essential results are presented and discussed in a clear and concise manner, highlighting their implications within the context of the study. We believe the current version of the discussion appropriately addresses the core objectives of the work.

Round 2
Reviewer 1 Report
Comments and Suggestions for Authors
The authors responded to all comments. However, in my opinion, a one-year study is preliminary, even if 13 experiments were conducted.
Agricultural research, after all, is highly variable.
Furthermore, in my opinion, we are talking about two experiments, with very extensive experimental variants.
What differentiates these experiments? The cultivars and the density, but I think we can consider this as a variants of the experiment.
Author Response
Responses to the reviewer:
QUESTIONS:
1 - The authors responded to all comments. However, in my opinion, a one-year study is preliminary, even if 13 experiments were conducted.
2 - Agricultural research, after all, is highly variable.
3 - Furthermore, in my opinion, we are talking about two experiments, with very extensive experimental variants. What differentiates these experiments? The cultivars and the density, but I think we can consider this as a variants of the experiment.
UNIFIED ANSWER:
We agree with the reviewer that a one-year study can be considered preliminary, especially under field conditions. However, in our experience, when experiments are conducted under controlled greenhouse conditions, repeating trials across years rarely produces substantially different results. For example, in previous studies with common bean × horseweed, canola × ryegrass/wild radish, and maize × pigweed, no significant year-to-year differences were observed. Variation, when it occurs, is mainly due to substrate characteristics, fertilization, liming, or seed lot quality. When these factors are standardized, coefficients of variation remain very low and differences between years become negligible. Therefore, we consider that repeating experiments in different years under protected conditions is not necessary to ensure robustness. Instead, the critical issue is to design experiments with sufficient statistical power to discriminate treatments (Concenço et al., 2018). If experimental errors or coefficients of variation indicate problems, the trial is repeated within the same season. This approach reduces time, material, and costs while maintaining reliability. In addition, the use of separate unifactorial experiments, rather than a single bifactorial design, allows for more accurate estimation of experimental error, higher statistical power, and more robust conclusions (Steel & Torrie, 1980).
The present study, conducted as a replacement series, followed established methodology (Radosevich, 1985; Cousens, 1991; Bianchi et al., 2006). In the first phase, we determined the plant density at which dry matter yield stabilizes, using monocultures of both crop and weed species (ranging from 1 to 64 plants per pot). Only after establishing this point did we proceed to the second phase, testing competition at fixed total densities but varying crop:weed proportions (e.g., 100:0, 75:25, 50:50, 25:75, 0:100). This design enables clear evaluation of crop–weed interactions without introducing confounding factors from unnecessary genotype comparisons. As a result, our trials provide a consistent and practical perspective on how different cultivars perform when associated with weed densities, while avoiding the excessive noise and reduced statistical power that would result from imposing a large bifactorial design.
REFERENCES:
BIANCHI M. A.; FLECK, N. G.; LAMEGO, F. P. Proportion among soybean and competitor plants and the relations of mutual interference. Ciência Rural, v. 36, n.5, p. 1380-1387, 2006. https://doi.org/10.1590/S0103-84782006000500006.
COUSENS, R. Aspects of the design and interpretation of competition (interference) experiments. Weed Technology, 5, 664-673, 1991. https://doi.org/10.1017/S0890037X00027524
CONCENÇO, G.; ANDRES, A.; SCHREIBER, F.; SCHERNER, A.; BEHENCK, J.P. Statistical approaches in weed research: choosing wisely. Weed Control Journal, v.17, n.1, p.45-58, 2018. http://dx.doi.org/10.7824/rbh.v17i1.536
RADOSEVICH, S.R. Methods to study interactions among crops and weeds. Weed Technology, v.1,n.3, p.190-198, 1987. https://doi.org/10.1017/S0890037X00029523.
STEEL, R.G.D.; TORRIE, J.H. Principles and procedures of statistics: a biometrical approach. 2nd ed. New York: McGraw-Hill, 1980. 633p.
VILÁ, M.; WILLIAMSON, M.; LONSDALE, M. Competition experiments on alien weeds with crops: lessons for measuring plant invasion impact? Biological Invasions, v. 6, n. 1, p. 59-69, 2004. https://doi.org/10.1023/B:BINV.0000010122.77024.8a.
